# Gut Microbiota across Normal Gestation and Gestational Diabetes Mellitus: A Cohort Analysis

**DOI:** 10.3390/metabo12090796

**Published:** 2022-08-26

**Authors:** Patricia M. Dualib, Carla R. Taddei, Gabriel Fernandes, Camila R. S. Carvalho, Luiz Gustavo Sparvoli, Isis T. Silva, Rosiane Mattar, Sandra R. G. Ferreira, Sergio A. Dib, Bianca de Almeida-Pititto

**Affiliations:** 1Department of Medicine, Escola Paulista de Medicina, Universidade Federal de São Paulo, Sena Madureira, 1500, Vila Clementino, São Paulo CEP 04021-001, Brazil; 2Department of Clinical and Toxicological Analysis and Obstetrics, School of Arts, Sciences and Humanities, Universidade de São Paulo (USP), Av. Prof. Lineu Prestes 580—Bloco 17, São Paulo CEP 05508-000, Brazil; 3DepaBiosystems Informatics and Genomics Group, Instituto René Rachou—Fiocruz Minas, Av. Augusto de Lima, 1714, Belo Horizonte CEP 30190-002, Brazil; 4Graduate Program in Endocrinology and Metabology, Universidade Federal de São Paulo, Rua Estado de Israel, nº 639, Vila Clementino, São Paulo CEP 04022-001, Brazil; 5Nutrition Course, Centro Universitário Estácio de Sá, Rua Erê, 207, Belo Horizonte CEP 30411-052, Brazil; 6Departament of Obstetrics, Escola Paulista de Medicina, Universidade Federal de São Paulo, R. Napoleão de Barros, 875—Vila Clementino, São Paulo CEP 04024-002, Brazil; 7Department of Epidemiology, Escola de Saúde Pública, Universidade de São Paulo, Av. Dr. Arnaldo, 715—Cerqueira César, São Paulo CEP 01246-904, Brazil; 8Department of Preventive Medicine, Escola Paulista de Medicina, Campus São Paulo, Universidade Federal de São Paulo, Rua Botucatu, n° 740, Vila Clementino, São Paulo CEP 04023-062, Brazil

**Keywords:** gestational diabetes mellitus, gut microbiota, obesity, pregnancy

## Abstract

The prevalence of gestational diabetes mellitus (GDM) is a global public health concern. The mechanism that leads to glucose tolerance beyond normal physiological levels to pathogenic conditions remains incompletely understood, and it is speculated that the maternal microbiome may play an important role. This study analyzes the gut microbiota composition in each trimester of weight-matched women with and without GDM and examines possible bacterial genera associations with GDM. This study followed 56 pregnant women with GDM and 59 without admitted to the outpatient clinic during their first/second or third trimester of gestation. They were submitted to a standardized questionnaire, dietary recalls, clinical examination, biological sample collection, and molecular profiling of fecal microbiota. Women with GDM were older and had a higher number of pregnancies than normal-tolerant ones. There was no difference in alpha diversity, and the groups did not differ regarding the overall microbiota structure. A higher abundance of *Bacteroides* in the GDM group was found. A positive correlation between *Christensenellaceae* and *Intestinobacter* abundances with one-hour post-challenge plasma glucose and a negative correlation between *Enterococcus* and two-hour plasma glucose levels were observed. *Bifidobacterium* and *Peptococcus* abundances were increased in the third gestational trimester for both groups. The gut microbiota composition was not dependent on the presence of GDM weight-matched women throughout gestation. However, some genera abundances showed associations with glucose metabolism. Our findings may therefore encourage a deeper understanding of physiological and pathophysiological changes in the microbiota throughout pregnancy, which could have further implications for diseases prevention.

## 1. Introduction

Gestational diabetes mellitus (GDM) is defined as hyperglycemia first diagnosed during pregnancy, which does not fulfill the criteria for overt diabetes and is the most frequent endocrine disorder in pregnancy [1]. It is estimated that in Brazil 18% of women develop GDM using the IADPSG criteria [2].

It is well-known that hyperglycemia during gestation confers significant risks for adverse outcomes for the mother and the offspring such as preeclampsia, premature birth, polyhydramnios, neonatal hypoglycemia, respiratory distress, and longer ICU stays [3]. In addition to increased gestational complications, GDM has been associated with increased cardiometabolic risk during the mother and offspring’s lifespan [4,5,6,7].

Underlying mechanisms of glucose metabolism disturbance during gestation remain incompletely understood. There is evidence that the maternal microbiota may play a role [8]. Associations of the gut microbiota composition with body weight and conditions of insulin resistance have been reported [9,10]. Some genera have been associated with type 2 diabetes, and some pathways have been proposed as gut microbiota modulates inflammation, interacts with dietary constituents, affects gut permeability, glucose and lipid metabolism, insulin sensitivity, and overall energy homeostasis in host [11]. As far as GDM is concerned, heterogeneous data linking the gut microbiota with GDM across the gestation trimesters have been found [12,13].

Even during normal gestation, changes in the gut microbiota of healthy pregnant women have been documented from the first to the third trimester [14]. Comparisons between gut microbiota characteristics of normoglycemic pregnant women and those with GDM have been reported recently [15,16,17,18]. However, these data are still scarce among populations, and confounders have limited interpretation of the findings. Adjustments for body adiposity, gestational stage, and therapeutic approach seem to be necessary.

Therefore, the objective of this study was to analyze the gut microbiota composition in each gestational trimester of weight-matched pregnant women with and without GDM and examine possible associations of bacterial genera with GDM.

## 2. Materials and Methods

### 2.1. Study Population and Design

From September 2018 to December 2019, all pregnant women attending the Normal Gestation Out-patient Clinic at the Obstetrics Division and Gestational Diabetes Out-patient Clinic at the Diabetes Center at the Federal University of Sao Paulo, Sao Paulo state, Brazil, were invited to participate in the present study. The institutional ethics committee of the Federal University of Sao Paulo approved the study (CAAE: 89108618.0.0000.5505), and all the participants signed a consent form [19].

Eligibility criteria were age ≥ 18 years, in any trimester of gestation, overweight or obese (overweight with a Body Mass Index (BMI) between 25 and 29.9 kg/m^2^ and obese with a BMI between 30 and 39.9 kg/m^2^), without known autoimmune diseases or chronic use of medications, particularly metformin, or inflammatory bowel disease. A total of 143 pregnant women were included in the study; 69 had GDM, and 74 normal-tolerant pregnant women were considered controls. For 115, stool samples were obtained during the gestational period (first or second trimester and/or third trimester). We excluded women who had used antibiotics, laxatives, or probiotics in the last 30 days to collect stool samples. After exclusion criteria, stool samples were available from 56 women with GDM and 59 without GDM.

IAPDSG criteria was used for GDM diagnosis (excluding women with overt diabetes), which is similar to the Brazilian guideline [20]. However, in this study, GDM as diagnosed when fasting plasma glucose was greater than 100 mg/dL in the first trimester or when at least two points were abnormal during the 75 g oral glucose tolerance (OGTT) (>92, >180, >153 mg/dL) in the third trimester. These criteria were employed to exclude borderline cases of GDM since, in previous studies, gut microbiota comparisons between normal pregnant women and those with mild hyperglycemia had differences in composition minimized [16,17].

The women were enrolled in the study in the first trimester (control = 54 and GDM = 36), and some in the third (control = 5 and GDM = 20); some mothers were diagnosed with gestational diabetes between 24 and 28 weeks of gestation. The study design was longitudinal, and, in each trimester, the participants were submitted to standardized questionnaires and clinical data collection.

### 2.2. Standardized Questionnaires

Through standardized questionnaires, completed under the supervision of trained interviewers, demographic, socioeconomic, lifestyle data, pre-pregnancy weight, and morbid personal and family history of the mother were obtained in the first and third trimesters of pregnancy.

### 2.3. Clinical Data

Pre-pregnancy weight was self-reported at the first visit to the clinics. Weight was obtained on a digital scale (Rice Lake, Sao Paulo) with an accuracy of 100 g and height precision of 0.5 cm, and these were used to calculate BMI. The neck circumference was measured with non-flexible tape (cm) immediately below the cricoid cartilage and perpendicular to the neck’s long axis, with the participant seated. Blood pressure was taken seated three times after a 5 min rest, using a mercury sphygmomanometer, with the cuff adjusted for the brachial circumference. Values considered were the arithmetic mean of the last two measurements.

### 2.4. Dietary Assessment

Participants were oriented to register all food and beverages consumed over three days, and to write down food consumed outside the home. A nutritionist showed how to record the information using traditional homemade utensils (cups, glasses, cutlery, and plates) and food models. Registers must be on alternate days and cover a weekend day [21]. The total energy, macro, and micronutrient intakes were calculated using Diet Pro software [22], using the Brazilian Food Composition Table (TACO) [23] as reference.

### 2.5. Stool Collection

The collection of feces was carried out at the house of the pregnant woman on the day before our evaluation. Fecal samples were collected in a standardized container supplied, and delivered during the visit to the clinic, stored for a maximum of 24 h at 2 to 5 °C. One researcher was responsible for following a standardized procedure (antiseptic handling, collection of aliquots in sterile cryotubes, and immediate freezing at 2–80). Aliquots were stored at −80 °C until DNA extraction. The stool samples were collected after the diagnosis of GDM.

### 2.6. Laboratory Tests

Routine prenatal care laboratory data, including fasting plasma glucose and lipid profile, were collected at entry and during the gestational period. Plasma glucose was determined by the glucose oxidase method. Concentrations of total cholesterol, HDL-c, and triglycerides were measured by enzymatic colorimetric methods, processed in an automatic analyzer. LDL-c and VLDL-c were calculated using the Friedewald equation.

### 2.7. Analytic Methods

Initially, bacterial DNA from stool samples was extracted using the QIAamp DNA Stool Mini Kit (Qiagen). The V4 region of the 16S rRNA gene was amplified by 25 cycles, using the previously described primers and conditions [24]. Negative controls with a buffer from the DNA extraction kit were included in the PCR runs. The amplicons were pooled and loaded on an Illumina MiSeq clamshell-style cartridge kit v2 at 500 cycles for paired-end 250 sequencing at a final concentration of 12 pM. The library was clustered to a density of approximately 820 k/mm^2^. Sequencing was based on a pool of 100 samples on two GS FLX Picotiter Plate plates, totaling two races. Thus, around 95,000 reads were obtained for each sample. The raw read files were processed in the R environment using the dada2 package [25]. During the process, the primers were removed, and the forward and reverse sequences were cut to 180 and 160 bases, respectively. Strings that contained more than two expected errors were removed. The filtered sequences had their errors corrected by the algorithm and were joined to form the ASVs (amplicon sequence variants). The chimeric sequences were also removed, and a sample count table was generated. The taxonomic classification was made using the tag.me package [26] using the model 515F-806R. The beta diversity was calculated using the Jensen–Shannon divergence matrix on PCoA analysis performed by the ade4 R package [27] for each library, and its analysis refers to the variety and complexity of species in a community. We used the Chao1 wealth estimate and Shannon and Simpson’s diversity indexes to calculate the alpha diversity. Alpha diversity is the total number of species in a habitat and beta diversity is the difference in species composition along an environmental gradient [28].

### 2.8. Statistical Analysis

Analyses were performed considering gestational trimester of collection and group of pregnant women stratified according to the presence of GDM.

Continuous variables were presented as mean (standard deviation) and categorical variables as frequency (percentage). Clinical and laboratory variables were compared by Student *t*-test and Mann–Whitney (continuous variables) or chi-squared tests (categorical variables). A *p*-value of <0.05 was considered statistically significant. Statistical Package for the Social Sciences^®^, version 22.0 was used. Microbiota analyses and graphics were performed using R software. For differences in the microbiota composition, PERMANOVA was performed for each site using Adonis function in vegan with Jensen–Shannon distances. For each variable, 999 permutations were made. The Wald test of the DESeq2 package [10.1186/s13059-014-0550-8] and an adjusted *p*-value filter of *p* < 0.01 were used to identify the differentially abundant genera. We used Spearman to see the correlation of bacterial genera with clinical and laboratory variables.

## 3. Results

The baseline characteristics of the participants in the cohort are in Table 1. Women with GDM were older, had a higher number of pregnancies, and tended to a higher frequency of family history of diabetes. On average, women from both groups were overweight or obese before pregnancy.

Fifty-four control women and 36 women with GDM were analyzed between the first and the second gestational trimester; GDM women had greater neck circumference (35.9 ± 2.9 vs. 34.4 ± 1.8, *p* < 0.01) and reported having had food counseling (*p* < 0.001) (Table 2).

In the third trimester, 55 control women and 54 GDM women were analyzed. The GDM group had higher mean values of neck circumference (36.4 ± 2.4 vs. 34.6 ± 2.1, *p* < 0.01), higher systolic blood pressure (117 ± 11 vs. 109 ± 10, *p* < 0.01) and reported more dietary counseling (*p* < 0.01) (Table 2). Gestational weight gain was similar in the two groups (*p* = 0.24). When we subdivided the groups into overweight and obesity, the gestational weight gain was 11.3 kgs (5.7) for the overweight control group and 10.2 kgs (5.7) for the overweight GDM group (*p* = 0.47), and 9.3 kgs (7.7) for the obese control group and 8.2 kgs (5.8) for the obese GDM group (*p* = 0.54); data not shown.

The dietary assessment showed that the control group consumed more carbohydrates (*p* = 0.01) and less protein (*p* = 0.00) in the first and second trimesters; in the third trimester, the control group consumed less protein also (*p* = 0.02), but the other differences disappeared (Table 2).

Regarding the routine exams performed during pregnancy, women with GDM had higher mean values of HbA1c, triglycerides, and plasma fasting blood glucose and at all points of the OGTT in the third trimester (Table 3).

### Microbiota Composition

We analyzed stool samples from 20 GDM women and 41 control women in the first/second trimester and from 49 GDM and 56 controls in the third trimester. Of these, 14 women with GDM and 34 control women were analyzed at both collection times (Figure 1).

There was no difference in alpha diversity between the control and the GDM groups. The overall structure of the microbiota did not differ, either (Figure 2, panel A). The comparison of microbiota composition between the groups across gestation showed no significant difference (Figure 2, panel B).

The comparisons of genera abundances verified a greater taxon of *Bifidobacterium* (*p* < 0.001 and LogFoldChange: 2.42) and *Peptococcus* (*p* < 0.001 and LogFoldChange: 2.74) in the third trimester (Figure 3, panel A) than in the first/second trimesters in both groups*. Bacteroides* genus was more abundant in women with GDM (*p* < 0.001 and LogFoldChange: 0.97) compared with those without GDM (Figure 3, panel B).

Correlations between fasting (in the beginning of pregnancy) and post-load plasma glucose (between 24th and 28th weeks of gestation) with the bacterial community structure were tested. A positive correlation between *Christensenellaceae* family (*p* < 0.001, rho: 0.37) and *Intestinobacter* (*p*: 0.02, rho: 0.32) with one-hour post-load plasma glucose level and a negative correlation between *Enterococcus* (*p*: 0.002, rho: −0.32) and two-hour post-load plasma glucose level were observed (Figure 4).

There are some correlations of bacterial genera with food intake, especially *Ruminococcus* with carbohydrate intake (rho: 0.39; *p*: 0.04) in the first and second trimesters (Appendix A).

## 4. Discussion

Our study had a unique opportunity to compare the gut microbiota composition in each gestational trimester of weight-matched Brazilian women with and without GDM and detected that GDM was not associated with changes in the overall microbiota structure throughout pregnancy. However, differences in the relative abundance of some bacteria in all women by gestational trimester and in those who developed GDM were detected, as well as correlations of some genera or families with plasma glucose levels.

The lack of difference in the overall structure of the microbiota (beta diversity) throughout the trimesters of pregnancy in our sample is similar to the findings of a study in which three periods (11–13, 23–28, and 33–38 weeks) were considered [29]; the alpha and beta diversity of the microbiota of the pregnant women remained unchanged during the entire pregnancy. Another study evaluated the gut microbiota assessed at 16 and 28-weeks’ gestation of overweight and obese women with no impaired glucose tolerance or impaired fasted and found results similar to ours; the relative abundances of key bacterial genera were not significantly altered between the groups, concluding that the gut microbiota does not likely have a disease specific characterization in GDM [30]. Contrasting findings have also been reported [31,32]. Koren et al. [14] found a reduction in diversity throughout gestation and increased *Proteobacteria* and *Actinobacteria*, but these changes were not related to the GDM occurrence. It must be considered whether there was a difference in BMI among the women, also differences in ethnicity, the country where the study was carried out, and socio-economic situation. The pregnant women evaluated in our study had similar BMI and were of low socioeconomic status. We found a positive correlation between *Ruminococcaceae*, a family of anaerobe bacteria in the class *Clostridia*, and carbohydrate intake in the first/second trimesters. Other studies also found results in this family of bacteria, such as a follow-up of 75 overweight or obese pregnant women in Finland, in which 15 changed their composition, but only the subset who developed GDM had an increased abundance in the *Ruminococcaceae,* a family of anaerobe bacteria in the class *Clostridia* [15]. A similar study, evaluating fecal microbiota profiles from overweight and obese pregnant women, showed that a high abundance of family *Ruminococcaceae* in early pregnancy might be related to adverse metabolic health [13]. The potential mechanism for these findings may be the impairment of glucose homoeostasis induced by the microbiota, since *Ruminococcaceae* was correlated with fasting glucose which, together with pregnancy, induced insulin resistance that likely contributes to the onset of GDM. *Ruminococcaceae* family has high metabolic activity including production of short-chain fatty acids and enhanced energy harvest [33].

Regarding relative abundance, we found that two bacteria were more relatively abundant in the third trimester, *Bifidobacterium* and *Peptococcus*, independent of GDM status. *Bifidobacterium* includes anaerobic bacteria representing one of the largest groups of bacteria that make up the intestinal microbiota and promotes health benefits to their hosts. A study showed that *Bifidobacterium* taxa was higher in normal-weight pregnant women than in overweight pregnant women, and a similar trend was seen in those with normal gestational weight gain compared with excessive weight gain over pregnancy [34]. Despite being overweight or obese, most pregnant women in our study had adequate gestational weight gain, according to the Institute of Medicine [35], and maybe a relatively higher abundance of genus Bifidobacterium could confer protection against excessive weight gain in our participants. As a matter of fact, Collado et al. [36] reported that the difference in *Bifidobacterium* genus numbers between the third and first trimesters of pregnancy correlates with normal weight gain over pregnancy, reinforcing a protective role for *Bifidobacterium. Peptococcus* is a genus of gram-positive, non-spore-forming, anaerobic or microaerophilic bacteria. In animals, prenatal stress was investigated in association with neurodevelopmental, cardiovascular, and metabolic disorders in the offspring [37]. Investigators showed that prenatally stressed animals increased the abundances of *Oscillibacter, Anaerotruncus*, and *Peptococcus* genera. Also, in humans, the last trimester of pregnant women could be considered the most stressful one.

As far as the occurrence of GDM is concerned, the alpha and beta diversities did not display differences between our GDM and non-GDM groups, although the relative abundance of specific bacteria did. Following our results, Cortez et al. [16] found no statistical difference in microbiota composition of 26 Brazilian women with GDM and 42 controls. Such concordance of studies including women with similar ancestry is relevant considering the role of environmental factors (mainly dietary habits) to the gut microbiota structure. A strength of ours was the inclusion of weight-matched women in the study groups, since an influence of BMI on microbiota composition had been described [38]. Obesity has been associated with intestinal dysbiosis [9] and is a recognized risk factor for GDM [39]. Contrasting findings among the reported studies could be explained, at least in part, by the difference in pregestational BMI of the participants. Such a potential confounder was minimized in our study. In addition, in an attempt to assure the GDM diagnosis, avoiding near-normal metabolic status, women were considered diabetic only when two or more points were altered in OGTT. Studies conducted in distinct populations reported differences in diversity and/or bacterial abundances in pregnant women who had or not GDM [31,32]. An increase in *Gammaproteobacteria* and *Haemophilus* abundances was described in Chinese women with GDM compared to normal-tolerant in a cross-sectional study [40].

We found a higher relative abundance of *Bacteroides* in pregnant women with GDM. This genus of gram-negative, anaerobic bacteria plays an important role in processing complex molecules to simpler ones in the intestine. In fact, it was reported that those who eat plenty of protein and animal fats have *Bacteroides* bacteria predominantly, while for those who consume more carbohydrates, the *Prevotella* species dominate [41]. GDM participants during the first/second trimester consumed fewer carbohydrates and more proteins, but similar ingestion of lipids. Associations of the gut microbiota with nutrient intakes and anthropometric and laboratory variables were previously investigated in 41 GDM women [42]. Those who were adherent to the dietary recommendations showed a better metabolic and inflammatory pattern at the end of the study with a significant decrease in *Bacteroides*. Even before pregnancy, our sample had excessive body adiposity. It was previously shown that prepregnancy obesity influenced circulating cytokines, chemokine, adipokines, and the gut microbiota, characterized by an increased abundance of *Bacteroides* [43].

Interesting correlations of bacteria and laboratory markers were found in our study, namely a positive correlation between *Christensenellaceae* and *Intestinobacter* with stimulated 1 h glucose level, and a negative correlation between Enterococcus and stimulated 2 h glucose level. Our finding is in agreement with a cross-sectional analysis of gut microbiota profile of healthy and GDM pregnant Danish women in the third trimester and eight months postpartum in whom the *Christensenella* genus was associated with increased fasting plasma glucose [17]. In other populations, this genus was inversely related to hosting BMI [44,45]. To the best of our knowledge, correlations of *Intestinibacter* in pregnant women have not been described. In a clinical trial conducted in 27 healthy young Danish men, metformin reduced abundance of the *Intestinibacter* spp. and *Clostridium* spp., and increased the abundance of *Escherichia/Shigella* spp. and *Bilophila wadsworthia* [46]. In another study in normal weight and obese children, *Intestinibacter*, among other genera, was associated with obesity and fasting insulin [47]. Animal models have been an important strategy in improving knowledge about the role of gut bacteria for some phenotypes such as obesity and diabetes. In this sense, *Bifidobacteria* and *Akkermansia muciniphila* were associated with favorable phenotypes [48], while interaction between *Clostridiales* and *Enterococcus* with caecal metabolism was proposed to play a role in the development of diabetes [49]. These correlations can identify potential preventive foci.

A recent systematic review involving 23 studies showed that there is a relationship between the intestinal microbiota and GDM, with the abundance of some bacteria related to altered glucose metabolism and less alpha and beta diversity and pointed out an important limitation: most of the studies included participants with different pre-gestational BMI [50].

The present study assessed dietary characteristics, but macronutrient intake did not help to explain differences in GDM occurrence nor microbiota composition. Some dietary habits, particularly a cafeteria pattern, were previously associated with a proinflammatory status and reduced insulin sensitivity [51]. Our findings did not support that the participants who developed GDM could be triggered partially by the diet adopted during pregnancy, despite having a lower energy and carbohydrate intake when assessed in the first trimester. It is known that long-term diet is strongly associated with the gut microbiome composition [52].

A mechanism linking changes in the gut microbiome with GDM involves the potential of gram-negative bacteria, like *Bacteroides* and *Christensenellaceae,* to induce metabolic endotoxemia. Two studies found an increase in pathways related to lipopolysaccharides (LPS) biosynthesis in women with GDM [42,53]. Increased intestinal permeability facilitated the LPS translocation into the circulation and triggered inflammation, which deteriorates insulin signaling in pregnant women, leading to glucose intolerance.

In our study, potential confounders in associations were minimized since participants were weight-matched and were not taking metformin [54,55]. Insulin was the only treatment prescribed for GDM patients in our clinics. In addition, the diets of both groups were similar during the follow-up.

Our study has limitations. All participants were overweight or obese, which impedes the extrapolation of results for other BMI categories. Investigations of pregnant women’s gut microbiota with excessive adiposity were reported but not including GDM [56]. We highlighted the strength of establishing GDM diagnosis only in women with two abnormal values in the OGTT and/or fasting glycemia above 100 mg/dL.

## 5. Conclusions

In summary, we conclude that in weight-matched Brazilian pregnant women, the gut microbiota composition may change in the last gestational trimester independent of the presence of GDM. This abnormality was not associated with changes in overall microbiome structure, but differences in the relative abundance of some bacteria (*Peptococcus* and *Bifidobacteria*) in all pregnant women can be detected, as well as in those who developed GDM (*Bacteroides*).

Our findings may contribute to furthering knowledge of the physiological and pathophysiological changes in the gut microbiota throughout pregnancy, which could have further implications for disease prevention.

## Figures and Tables

**Figure 1 metabolites-12-00796-f001:**
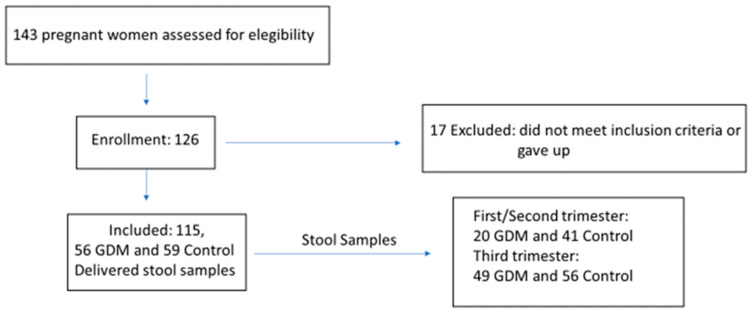
Study flowchart.

**Figure 2 metabolites-12-00796-f002:**
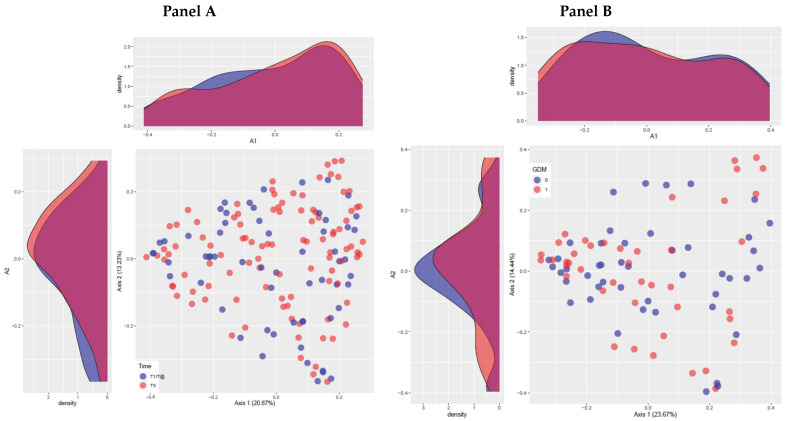
Microbiota structure of the total sample (panel (**A**)) and groups of normal and GDM participants (panel (**B**)) across gestation. Axes percentages represent the amount of variation in the data explained by the axis, calculated from the eigenvalues of PCoA (main coordinate analysis). The axes intervals represent the relative dissimilarity present between the samples. Time, trimester of pregnancy: T1/2, first or second trimesters of gestation; T3, third trimester of gestation. GDM: 0, without GDM; 1, with GDM.

**Figure 3 metabolites-12-00796-f003:**
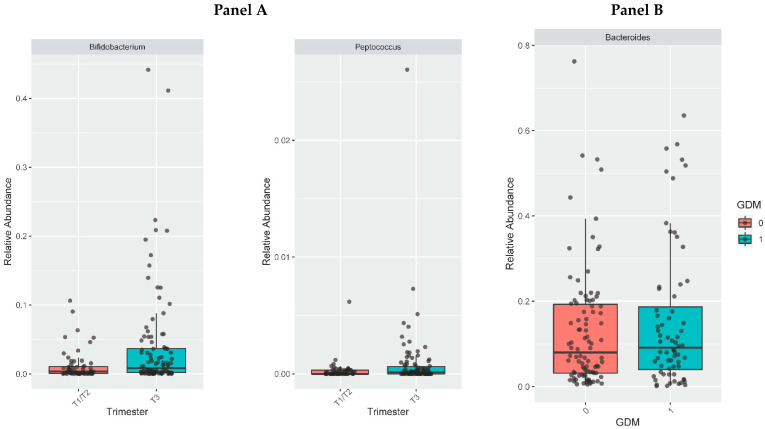
Abundance of selected genera according to trimester of pregnancy (panel (**A**)) and presence of gestational diabetes (panel (**B**)). Panel (**A**): abundance of *Bifidobacterium* and *Peptococcus*. Panel (**B**): abundance of *Bacteroides*.

**Figure 4 metabolites-12-00796-f004:**
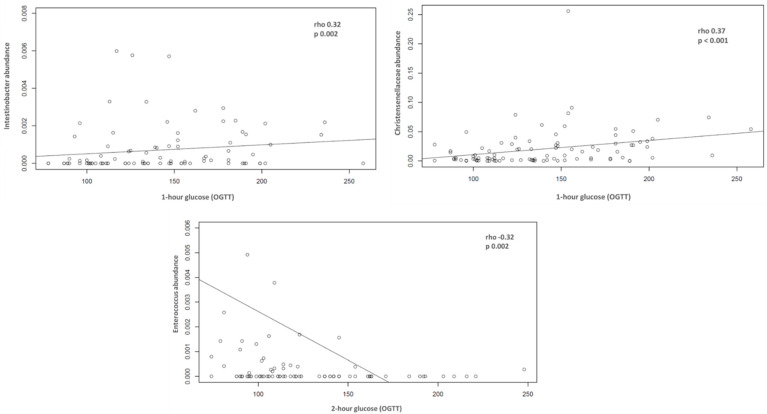
Correlation between bacteria and fasting and post load plasma glucose in the third trimester of pregnancy.

**Table 1 metabolites-12-00796-t001:** Baseline characteristics of pregnant women with or without GDM.

Characteristics	Control (*n* = 59)	GDM (*n* = 56)	*p*
**Age (years)**	28.1 (5.9) ^α^	33.2 (6.2) ^α^	<0.01
**Pregestational BMI (kg/m^2^)**	29.2 (3.7) ^α^	30.2 (3.9) ^α^	0.39
**Race, *n* (%)**			0.73
Caucasian	25 (42.4)	22 (39.3)	
Black	12 (20.4)	10 (17.9)	
Mulatto	22 (37.2)	24 (42.8)	
**Schooling, *n* (%)**			0.22
Up to 7 years	1 (1.7)	7 (12.5)	
to 13 years	43 (76.3)	37 66.1)	
≥14 years	13 (22.0)	12 (21.4)	
**Number of pregnancies**			0.01
1 pregnancy	24 (40.6)	9 (16.1)	
2 pregnancies	17 (28.8)	15 (26.8)	
3 or more pregnancies	18 (30.6)	32 (57.1)	
**Family History of diabetes, *n* (%)**	16 (27.1)	24 (43.6)	0.07
**Gestational weight gain (kgs)**	10.7 (6.3) ^α^	9.2 (5.8) ^α^	0.24
**Pregestational physical activity ** **≥ 150 min per week, *n* (%)**	18 (30.0)	18 (32.0)	0.36

*n*: number of pregnant women. Student *t*-test (continuous variables) or chi-squared (qualitative variables) were used. Values are mean (SD) ^α^ or *n* (%).

**Table 2 metabolites-12-00796-t002:** Clinical characteristics and behavioral habits of pregnant women with or without GDM during the pregnancy.

Clinical Characteristics and Dietary Data	First/Second Trimester			Third Trimester		
	Control (*n* = 54)	GDM (*n* = 36)	*p*	Control (*n* = 55)	GDM (*n* = 54)	*p*
**Gestational age(weeks)**	19.4 (4.2) ^α^	19.0 (5.2) ^α^	0.71	33.5 (2.4) ^α^	32.9 (2.3) ^α^	0.25
**Physical activity (%)**	27 (49.1)	22 (62.9)	0.20	25 (46.3)	24.0 (53.8)	0.99
**Smoking (%)**	0	3 (8.6)	0.27	0	0	--
**Alcohol consumption (%)**	2 (3.6)	1 (2.9)	0.84	0	0	--
**Dietary orientation (%)**	22 (40.7)	36 (100)	<0.01	24 (45.3)	52.0 (100)	<0.01
**Neck circumference(cm)**	34.5 (1.8) ^α^	35.9 (2.9 ^α^)	<0.01	34.6 (2.1) ^α^	36.5 (2.4) ^α^	<0.01
**Systolic blood pressure (mmHg)**	109 (12.4) ^α^	112 (11.4) ^α^	0.25	108 (9.9) ^α^	117 (11.6) ^α^	<0.01
**Diastolic blood pressure (mmHg)**	68 (10.4) ^α^	69 (9.1) ^α^	0.49	67 (9.1) ^α^	72 (11.1) ^α^	0.09
**Dietary data ^#^**	*n* = 40	*n* = 18		*n* = 48	*n* = 44	
Energy intake (kcals)	1940 (1385–2504)	1294 (1145–1880)	0.01	1789 (1439–2137)	1654 (1479–1942)	0.49
Carbohydrates (%TEI)	47.7 (43.2–57.5)	40.3 (34.7–49.2)	0.01	48.6 (42.4–57.6)	45.1 (40.9–54.8)	0.14
Proteins (%TEI)	13.1 (11.3–17.5)	24.8 (18.9–27.8)	0.00	15.6 (11.4–20.3)	18.7 (15.4–25.9)	0.02
Lipids (%TEI)	37.2 (28.4–41.3)	33.7 (27.0–40.3)	0.73	32.9 (28.5–39.9)	33.9 (29.5–38.5)	0.74
Saturated Fat (% TEI)	10.5 (6.9–13.8)	8.6 (6.5–10.9)	0.47	8.4 (6.3–11.5)	9.3 (5.7–11.0)	0.96
Monounsaturated Fat (% TEI)	8.9 (6.3–11.8)	8.6 (6.2–13.7)	0.83	8.6 (6.4–12.1)	9.9 (6.1–11.7)	0.71
Polyunsaturated Fat (% TEI)	7.2 (5.8–10.3)	9.4 (7.8–11.7)	0.07	8.1 (5.6–10.3)	8.6 (6.6–10.6)	0.33
Total Fiber (g)	7.8 (5.2–15.3)	11.8 (8.5–13.4)	0.17	8.9 (5.6–13.4)	8.6 (6.2–13.4)	0.87

Student *t*-test and Mann–Whitney # (continuous variables) or chi-squared (qualitative variables) were used. Values are mean (SD) ^α^ or (%) or median values (IIQ). TEI: total energy intake. *n*: number of pregnant women.

**Table 3 metabolites-12-00796-t003:** Laboratory data obtained in the third trimester of pregnancy of women without and with GDM.

Laboratory Data	Control (*n* = 60)	GDM (*n* = 56)	*p*
**Fasting Blood Glucose (mg/dL)**	85.2 (5.8)	99.3 (12.6)	<0.01
**OGTT:**			
Fasting (mg/dL)	81.4 (12.4)	99.3 (21.9)	<0.01
1 h (mg/dL)	126.4 (27.0)	185.8 (33.1)	<0.01
2 h (mg/dL)	109.9 (20.2)	169.9 (38.4)	<0.01
**HbA1c (%)**	5.2(0.1)	5.7 (0.5)	0.03
**Total Cholesterol (mg/dL)**	185.2 (33.7)	206 (44.8)	0.18
**HDL Cholesterol (mg/dL)**	63.8 (14.6)	62.0 (15.2)	0.73
**LDL Cholesterol (mg/dL)**	96.3 (29.6)	114.2 (36.6)	0.16
**Triglycerides (mg/dL)**	133.0 (53.9)	196.8 (68.6)	0.01

OGTT: oral glucose tolerance test. Student *t*-test (continuous variables) was used. Values are mean (SD). *n*: number of pregnant women.

## Data Availability

The data presented in this study are available on request from the corresponding author. The data are not publicly available due to not been published to a public repository.

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
