# Peer review of "Gut Microbiota across Normal Gestation and Gestational Diabetes Mellitus: A Cohort Analysis"

_metabolites, 2022, doi:10.3390/metabo12090796_

Round 1
Reviewer 1 Report
This manuscript is about the role of maternal microbiome in gestational diabetes mellitus. The authors have analyzed the gut microbiota composition in each trimester of weight-matched women with and without gestational diabetes mellitus. They have examined possible bacterial genera associations with gestational diabetes mellitus. The manuscript is well written. There was no differences between groups regarding the overall microbiota structure, neither in alpha diversity. The authors concluded that the gut microbiota composition was not dependent on the presence of gestational diabetes mellitus weight-matched women throughout gestation. It was already shown that gut microbiota does not likely have a disease specific characterization in gestational diabetes mellitus. Therefore, the results are not surprinsing. However, an important point of this study, according with the authors, is the correlation of Intestinibacter in pregnant women.
There are some issues and therefore, I have some comments.
Major revision
Long-term diet is strongly associated with the gut microbiome composition. How do you explain that the findings of your study did not support that the participants who developed GDM could be trigger partially by the diet adopted during pregnancy, despite having a lower energy and carbohydrate intake when assessed in the first trimester?
Page 9 line 285: The authors have made a speculation: ``a relatively higher abundance of this genus could confer protection against excessive weight gain.`` It could be dangerous this kind of speculation, apart from the data from literature. The same for line 296: `` in humans, the last trimester of pregnant women could be considered the most stressful one.``
Why do you not put all figures in manuscript and you consider putting in supplementary files? There are only 2 figures in the results of this manuscript.
What is the explanation of the authors for this result (the gut microbiota composition was not dependent on the presence of gestational diabetes mellitus) in spite of the literature’s data?
Please put in discussion, all data literature about the subject of this study in a table (with: author and year, study theme, methods, results) only for gestational diabetes mellitus. In addition, I recommend making before conclusion, a synthesis paragraph with known and new data of this article, for increasing the quality of this study.
Minor revision
Font size in tables are too big.
Reviewer 2 Report
This study analyzes the gut microbiota composition in each trimester of weight-matched women with and without GDM and examines possible bacterial genera associations with GDM. In general, this manuscript is fine, the authors showed evidence to support their overall conclusions. However, some problems needed to revise, the detail comments are as followed:
1. Stool collection:In the method, it is required to specify the collection time of the patient's feces (in the morning)? The site selected for fecal collection (all, middle or end)?
2. In the method, the reference criteria for over- weight or obese should be described or explained.
3. Table 1.:to 13 years “43 76.3) 37 66.1)”?
4. Some of the indicators in Table 1-3 are percentages, some are composition ratios,and some SD which should be clearly marked using *. In addition, does the percentage of Race add up to 100%?
5. The title of the picture should be under the picture, and all pictures need to be modified.
6. The picture definition is too low, please replace the picture (Figure 2-2. and Supplementary Figure S1.-S2.)
7. Why merged T1 and T2 be compared with T3 instead of each other?
Round 2
Reviewer 1 Report
Thank you for your responses.